# The Effects of Cellular Membrane Damage on the Long-Term Storage and Adhesion of Probiotic Bacteria in Caco-2 Cell Line

**DOI:** 10.3390/nu15153484

**Published:** 2023-08-07

**Authors:** Jakub Kiepś, Wojciech Juzwa, Anna Olejnik, Anna Sip, Jolanta Tomaszewska-Gras, Radosław Dembczyński

**Affiliations:** 1Department of Biotechnology and Food Microbiology, Poznan University of Life Sciences, 60-627 Poznan, Poland; wojciech.juzwa@up.poznan.pl (W.J.); anna.olejnik@up.poznan.pl (A.O.); anna.sip@up.poznan.pl (A.S.); 2Department of Food Safety and Quality Management, Poznan University of Life Sciences, 60-624 Poznan, Poland; jolanta.tomaszewska-gras@up.poznan.pl

**Keywords:** viability, fluid bed drying, lactic acid bacteria, stress, quality control, imaging flow cytometry

## Abstract

Adhesion is one of the main factors responsible for the probiotic properties of bacteria in the human gut. Membrane proteins affected by cellular damage are one of the key aspects determining adhesion. Fluid-bed-dried preparations containing probiotic bacteria were analyzed in terms of their stability (temperature of glass transition) and shelf life in different conditions (modified atmosphere, refrigeration). Imaging flow cytometry was utilized to determine four subpopulations of cells based on their physiological and morphological properties. Lastly, adhesion was measured in bacteria cultured in optimal conditions and treated with heat shock. The results show that the subpopulations with no or low levels of cell membrane damage exhibit the ability to adhere to Caco-2 cells. The temperature of protein denaturation in bacteria was recorded as being between 65 °C and 70 °C. The highest glass transition temperature (Tg) value for hydroxypropyl methylcellulose (used as a coating substance) was measured at 152.6 °C. Drying and coating can be utilized as a sufficient treatment, allowing a long shelf-life (up to 12 months). It is, however, worth noting that technological processing, especially with high temperatures, may decrease the probiotic value of the preparation by damaging the bacterial cells.

## 1. Introduction

Methods of microorganism preservation such as spray drying, freeze drying, vacuum drying, and fluidized bed drying allow probiotics to be obtained that retain their properties during storage and are easy to dose and apply [1]. The particular interest in probiotics stems from their beneficial effects on the health of humans and farm animals. Clinical trials confirm the beneficial effects of probiotics in the treatment of gastrointestinal conditions (GI) such as diarrhea, irritable bowel syndrome and inflammatory bowel disease [2]. In addition, their application in non-GI medical conditions is currently being researched, for example, in patients with atopic dermatitis and type-2 diabetes [3,4]. 

To obtain dried preparations of high quality and viability, various protection strategies can be incorporated into processing steps. In general, three strategies can be distinguished: adding protective agents, optimizing drying parameters, and prestressing the probiotic cells before drying [5]. The intentional use of stresses may increase the resistance of cells to unfavorable conditions during drying (dehydration, osmotic stress, shear stress, elevated temperature). All these stress conditions adversely affect the cells. As a result, damaged and dead cells appear in the preparation, reducing its quality (long-term stability and shelf-life) and affecting some of the probiotic properties (e.g., the ability to adhere to the intestinal epithelium). 

Adhesion can be analyzed using various in vitro and in vivo methods. In vitro models include cell cultures (most notably Caco-2 and HT-29 cell lines), intestinal mucus cell models, organ culture models, and whole tissue models. Due to their ease of application and their relatively low cost, these models are used as a foundation for adhesion tests. They can be analyzed using simple imaging techniques like SEM. Additionally, the development of new tissue and organ models makes it possible to better represent the structural architecture of the intestinal tissues using 3d designs and allows for obtaining a multilayered system incorporating all types of cells found in the intestines. Their main disadvantage is that they do not represent the specific physiology of the host and do not take into consideration the intestinal microbiota [6]. To overcome these issues, ex vivo and in vivo models have been utilized. The intestinal tissue can also be analyzed afterward using microbiological and molecular analysis. Genomics is a non-invasive method that has also been utilized to identify the proteins associated with adhesion [7].

To evaluate the viability of probiotic cells, the most commonly used method is pour plate counts. This makes it possible to enumerate the viable probiotic cells; however, it is limited only to the subpopulation of cells with the ability to grow and divide. Inactivated, killed, or dead cells, which are unrecognizable using classical microbiological methods, also possess functional properties; however, live cells are more efficacious [8]. Treatment with sublethal temperatures can induce cross-protection mechanisms in probiotic bacteria. The exposure to sublethal temperatures can trigger the synthesis of heat shock proteins and other stress response elements, which not only protect probiotic cells from temperature stress, but also confer enhanced resistance to other stresses, such as those encountered in the gut environment, such as low pH [9], as well as those encountered during the treatment process, such as high temperature and dehydration [10]. As stated by Wang et al. [11], the live/dead state does not influence the adhesion ability of certain probiotic strains. Adhesion is mediated mainly by the components on the cell surface; it is, however, correlated with the integrity of the cell membrane. The health benefits conferred by live cells are more complex than those of dead cells [8]. 

This study aims to select fluid bed drying parameters (such as the application of the coating step and the selection of the coating material) for selected probiotic bacteria strains, as well as to assess the impact of culture conditions and drying processes on the survival, storage stability, and physiological characteristics of probiotic bacteria cells. Thermal analyses (DSC and TG/DTA) are utilized to assess the glass transition temperature of different coating substances applied in the process. The effect of technological parameters (temperature) on cell mortality was also determined using DSC analysis. An assessment of the impact of the drying method and drying conditions used on cell survival was performed using a cell line to model bacterial adhesion to the intestinal epithelium in vitro. Imaging flow cytometry (IFC) was applied to analyze cell adhesion and assess the physiological and morphological properties of the cell. The main impact and the significant novelty of the employed methodology lie in the prospect of facilitating the detection of dormant and injured bacterial cells as separate subpopulations. This makes it possible to determine more subpopulations than the limited routine discrimination of live and dead cells only; in addition to these two subpopulations, we also determined two subpopulations with varying levels of membrane damage and ongoing metabolic activity. Our research is focused on the assessment of cell cellular damage and metabolic activity in the samples after adhesion in cell lines and after long-term storage, and showcases a novel way of assessing the viability of probiotic bacteria. The adhesion results were also determined using the conventional pour plate method.

## 2. Materials and Methods

### 2.1. Inoculum Preparation

*Leuconostoc mesenteroides* 51 KBiMŻ, *Enterococcus faecium* 73 KBiMŻ, and *Carnobacterium divergens* 3 KbiMŻ were chosen as strains for assessment and were stored in the form of beads with cells in a freezer at −80 °C. MRS broth was used for propagation as a medium ensuring optimal growth conditions for lactic acid bacteria. Inoculum was prepared in a volume of 10% of the bioreactor vessel to obtain the highest biomass yield. Two-step propagation also allowed for better adaptation of the microorganisms and shortened the resting phase in culture. The first step was carried out on ice to limit cell damage, which can result from thawing at room temperature. Afterward, the microorganisms were transferred to a 15 mL Falcon conical tube containing 9 mL MRS broth and sealed with parafilm, and incubated for 24 h at 30 °C. After incubation, 10 mL of the inoculum was transferred to a flask containing 90 mL of MRS broth, followed by another 24 h incubation at 30 °C. After these steps, the inoculum was ready to be used to start a culture in the bioreactor. All actions on microorganisms were carried out in a laminar chamber to reduce the risk of infection.

### 2.2. Bacterial Cultures

Biostat A plus bioreactors (Sartorius, Göttingen, Germany) were used for the cell culturing step. Firstly, the bioreactors and equipment were prepared: the pH electrode was calibrated against buffers at pH 4 and pH 9; next, 1 L of MRS broth medium was added to the vessel and then the bioreactor was autoclaved at 121 °C with a 20 min exposure time to provide sterile conditions. Inoculum was added, in a volume of 10% of the volume of the medium (100 mL), to a bioreactor cooled to 30 °C after sterilization. During this step, the bioreactor was pressurized with nitrogen to minimize the risk of infection with airborne microorganisms. After inoculation, the growth parameters were set at 30 °C, with a stirrer speed of 150 RPM and pH set at 6.5 (the optimal value for the selected strains). A 30% NaOH solution was used for pH regulation. NaOH consumption and pH changes over time were constantly monitored to determine the end of the exponential growth. The stabilization of pH at the set level with the simultaneous absence of base consumption indicated the end of microbial growth and inhibition of the production of acidifying metabolites. Heat shock stress was induced in certain samples by heating the culture in its stationary phase to 60 °C for 30 min. Samples were collected into sterile test tubes using a peristaltic pump connected to the sampling port.

### 2.3. Fluid Bed Drying

For the drying process, a GEA Strea-1 laboratory fluid-bed dryer was used. In the first step, the drum of the fluid-bed dryer was filled with the matrix (crystalline microcellulose). Next, the stream of filtered and heated air was introduced through the bottom perforated plate to keep the matrix in the fluid phase and to ensure even drying. The temperature range was set at up to 50 °C. Probiotic cells were suspended in the solution of a protective substance (5% trehalose) and fed to the dryer. They were then sprayed using an atomizing nozzle under a pressure of 2 bar. Drying and coating took approximately 30 min for each step. 

### 2.4. Plate Count Method

The plate counting method was used to determine the number of live microorganisms present in the analyzed samples after drying and during the analysis of adhesion. Tubes containing 9 mL of solvent (0.9% NaCl) were prepared, and then 1 mL of the sample was added into the first tube so that a dilution of 10^−1^ was obtained. After thorough vortexing, 1 mL of the sample with a dilution of 10^−1^ was transferred to the next tube, and the whole process was repeated until the required order of dilution was obtained. For the culture, the fluid and suspension for drying the samples were taken directly from the suspension; however, in the case of the dried formulation, it was necessary to add a rehydration step. For that purpose, 1 g of the sample was weighed into a 99 mL flask with 0.9% NaCl, which was then placed in a 37 °C water bath. After 30 min of shaking in a water bath, further dilutions were prepared using the suspension. Finally, 1 mL of the diluted sample was applied to Petri dishes, and MRS agar medium (previously sterilized and stored at 55 °C to prevent solidification) was poured over the samples, mixed, and left to solidify. The Petri dishes were incubated under aerobic conditions in an incubation chamber at 30 °C for 48 h, after which time visible colonies were counted and evaluated. The results obtained by counting visible colonies were considered statistically significant only for plates with dilution samples for which the number of colonies ranged from 30 to 300.

### 2.5. Intestinal Epithelial Cell Culture

The human intestinal epithelial Caco-2 cell line (HTB-37™) was obtained from American Type Culture Collection (ATCC, Manassas, VA, USA). Cells were cultured in Dulbecco’s Modified Eagle’s Medium (DMEM, Sigma-Aldrich, Saint Louis, MO, USA) supplemented with 1% non-essential amino acids (100X NEAA, Sigma-Aldrich) and 20% fetal bovine serum (FBS, Gibco BRL, Grand Island, NY, USA) and maintained at 37 °C in a humidified atmosphere of 95% air and 5% CO_2_. For intestinal barrier formation, Caco-2 cells were seeded on the PET membranes (Millicell^®^ Cell Culture Inserts, 24 mm diameter, 0.4 µm pore size) (Millipore, Burlington, MA, USA, Merck Group) at an initial density of 4 × 10^5^ cells/cm^2^ and cultured for 21 days with a medium change three times a week. The integrity of the Caco-2 cell monolayers was monitored on the basis of transepithelial electrical resistance (TEER) measurements using the Millicell Electrical Resistance System (ERS-2, Millipore). Caco-2 cell cultures with TEER values ≥ 600 Ω × cm^2^ were used in the bacteria adhesion experiments. 

Caco-2 is an epithelial cell line isolated from colon adenocarcinoma. One of the unique properties of the Caco-2 cell line is its ability to form a brush border with microvilli. Cells were cultivated for 21 days in 6-well plates with 0.4 µm cell culture inserts to reach the best divergence and full confluence. As a medium, D-MEM (Dulbecco’s modified Eagle medium) was chosen, with the addition of an antibiotic (gentamicin). Cell lines were cultivated under the following conditions: temperature of 37 °C in a controlled atmosphere with 5% CO_2_ concentration, and humidity exceeding 95%.

### 2.6. Adhesion Assay

Before the adhesion assay, Caco-2 cell monolayers were washed twice with PBS. Then, DMEM (without phenol red) with bacterial cells was added. The Caco-2 cell cultures combined with bacteria were incubated for 2 h at 37 °C. After incubation, the medium was removed from the epithelial cell cultures, and the cell monolayers were washed gently with PBS. Then, a cold 1% Triton X-100 solution was used to lyse the Caco-2 cells and release the adhered bacterial cells. Lysis was carried out for 3–5 min on ice. The cell lysates were centrifuged (10 min, 3.500 rpm), and the pellets were suspended in PBS. The number of bacterial cells was determined using IFC (Imaging Flow Cytometry) and pour plate counts in three replicates. Pour plate counts were performed in MRS agar and incubated under anaerobic conditions at 37 °C for 48 h.

### 2.7. Imaging Flow Cytometry

Imaging flow cytometry is a type of flow cytometry that combines the high-throughput analysis capabilities of flow cytometry with the imaging capabilities of microscopy. This method allows for the simultaneous analysis of multiple parameters, such as metabolic activity and cellular membrane integrity, in a single cell. The physiological and morphological properties of bacterial cells, such as metabolic activity and the integrity of the cellular membrane, were assessed using the imaging flow cytometer Amnis FlowSight™ (Luminex Corp., Austin, TX, USA). This imaging flow cytometer is equipped with 3 lasers (405 nm, 488 nm, and 642 nm), 5 fluorescence channels (acquisition by a multi-channel CCD camera), and a side scatter detector (SSC). Post-acquisition data analysis was performed using the IDEAS software ver. 6.2 (Luminex Corp., Austin, TX, USA). Morphological characteristics of the analyzed cells were determined using the Gradient RMS parameter from brightfield signals (Ch01) for the discrimination of high-resolution cell images. Additionally, brightfield digital image processing parameters: Aspect Ratio and Area were used to characterize the shape and size of the analyzed bacterial cells in combination with the discrimination of bacterial cells from non-cellular debris (particles from prebiotic component) and single cells from aggregates. The viability and activity of probiotic bacteria cells in the samples were determined by performing fluorescent staining with RedoxSensorTM Green and PI (propidium iodide). The samples for analysis were prepared by centrifugation and then suspended in a 1% PBS buffer in 1:200 dilution. Then, the following dyes were added to the samples with a volume of 500 µL by being pipetted into Eppendorf tubes: 1.6 µL of RedoxSensorTM Green and 1.2 µL of PI. The cells in the samples were counted and assessed for morphology (microscopic image), activity (signal for RedoxSensorTM Green), and integrity of the cell membrane (signal for PI). Additional dying with DRAQ 5 and wheat germ agglutinin (WGA) was used to show the pattern of adhesion of the bacterial cells onto the Caco-2 cells. DRAQ 5 was added to dye the epithelial cell nuclei red and WGA to dye the cell membranes of the bacteria green. For all samples, a previously developed machine learning (ML) protocol [12] was utilized to help in the discrimination of cells from cellular and non-cellular debris. The ML module is a part of the IDEAS^®^ 6.3. software, which was designed to process data acquired by Amnis Flow Sight imaging flow cytometer (Luminex Corp., Austin, TX, USA). 

### 2.8. Scanning Electron Microscopy (SEM)

To visualize the morphology of the dried powder microparticles, selected samples were analyzed using scanning electron microscopy. Firstly, samples of the powder were coated with a thin layer of gold using the Q15OT ES coater. Images were obtained using a Quanta 250 microscope.

### 2.9. Thermogravimetry–Differential Thermal Analysis (TG/DTA)

The TG/DTA analysis was conducted using an STA 449 F5 Jupiter (Netzsch, Selb, Germany). First, 30 mg of sample was placed in the heating chamber and heated to 200 °C at the rate of 5 °C/min. Then, the chamber was filled with an inert gas (helium), and its flow was set at 20 mL/min. 

### 2.10. Differential Scanning Calorimetry (DSC)

A differential scanning calorimeter DSC 8500 (PerkinElmer Inc., Waltham, MA, USA) was used to determine the glass transition phenomena. The device, which was equipped with an Intracooler II and was running under Pyris 10.1 instrument management software, was calibrated using the standards of indium (T_m_ = 156.60 °C, ΔH = 28.45 J/g, PerkinElmer Inc.) and n-dodecane (99.8 purity, T_m_ =−9.65 °C, Merck). The samples (approximately 5–6 mg) were weighed into 20 µL aluminum pans (PerkinElmer, No. 0219–0062, Waltham, MA, USA) and hermetically sealed. The analysis of glass transition involved the following steps: (1) holding for 1.0 min at 30 °C; and (2) heating from 30 °C to 300 °C at 5 °C/min. The reference was an empty, hermetically sealed aluminum pan. Glass transition, as a second-order phase transition, was identified by a step in the baseline of the measurement curve and registered as a heat capacity change (ΔC_p_, J/g °C) as a function of temperature. The glass transition temperature (T_g,_ °C) parameter was calculated as the inflection point. All samples were analyzed in two replicates.

### 2.11. Storage and Shelf-Life Tests

Samples after drying and coating were transferred to glass vials. A 1 g sample was placed in each vial and sealed using airtight caps in one of three variants: with atmospheric air, with nitrogen, and under vacuum. After sealing, the vials were stored at 3 different temperatures—−20 °C, 4 °C, and 20 °C—for 12 months. Samples for the pour plate tests were taken after 1, 2, 3, 6, 9, and 12 months of storage.

## 3. Results and Discussion

### 3.1. Glass Transition Temperature

The glass transition temperature was analyzed using both DSC and TG/DTA. This parameter is a property of amorphous materials that are formed, e.g., by removing the dispersing medium. This phenomenon occurs in the process of fluidized bed drying, where water (the dispersing medium) is removed and the duration of the process is insufficient for crystallization to occur; therefore, the dried material stays in an amorphous state. Above a critical temperature, described as the glass transition temperature, the dried material will start to change its structure from a glassy solid state to a rubbery form. This structural change can impact the physiochemical properties of a product and impact the viability of dried probiotics. Differential scanning calorimetry is a method that is well suited for measuring such changes in biological systems. It works by heating the sample with reference to an inert material (i.e., a material that is not undergoing a phase transition in a selected temperature range). Phase transitions such as glass transition are then registered as a difference in heat as a function of temperature between the sample and the reference material. The glass transition temperatures were measured for three dried samples coated with different coating materials (gum Arabic, hydroxypropyl methylcellulose, and shellac), as well as for dried, uncoated samples (Figure 1). All temperatures were high enough to provide stability and long shelf-life, and they are compared with the temperatures obtained using TG/DTA in Table 1. TG/DTA analysis was used to compare the T_g_ values. Figure 2 shows the recorded thermal properties of the samples, where it can be seen that the glass transition was recorded in a similar range as when using DSC. Samples after coating were also visualized using SEM (Figure 3) to show the surface of the pellets after using different coating materials. Additionally, bacterial cells immobilized on the surface of a single microcellulose pellet were observed (Figure 4).

### 3.2. Cells Pre Adhesion

Samples were taken pre adhesion from the suspension of probiotic cells used for incubation with epithelial cells. The following subpopulations were distinguished in the pre-adhesion samples using flow cytometry with RSG and PI staining: active (with confirmed metabolic activity and no cellular membrane damage); mid-active I (with lower-than-average levels of both metabolic activity and cellular membrane damage); mid-active II (with higher-than-average levels of both metabolic activity and cellular membrane damage) and dead (with no metabolic activity and high levels of cellular membrane damage). Detailed information on subpopulation determination was provided in our previous work [12]. The distribution of these groups for samples containing three strains of probiotic bacteria cultured under optimal conditions and after heat shock is presented in Figure 5, and the numbers of cells counted using the pour plate method is shown in Table 2. It can be observed that the majority of almost all samples corresponded to active cells, with the exception of *L. mesenteroides* after heat shock, which was more susceptible to high temperature and consisted mainly of dead cells, with almost equal amounts of active and mid-active II subpopulations. 

### 3.3. Cells Post Adhesion

To measure the distribution of subpopulations in the post-adhesion samples, the supernatant left after incubation and washing of the cells was analyzed. This was necessary since the samples after trypsinization contained a combination of bacterial and epithelial cells, and as such were not suitable for flow cytometry. They were still enumerated using the pour plate method. The post-adhesion samples showed very low numbers of active cells. This led to the conclusion that the active subpopulation was adhering to the Caco-2 cells during incubation. The composition of the samples (Figure 6), which exhibited higher contents of dead, mid-active I, and mid-active II cells than pre adhesion, showed that these subpopulations were unable to adhere, and were left suspended in the medium, which was analyzed for its contents. Caco-2 cells are a well-differentiated adherent cell line that is used for testing the adhesion of intestinal bacteria, studies of transmembrane transport, and studies of bacterial pathogen invasion. The results clearly show that the damage sustained by bacterial cells influences their ability to adhere to Caco-2 cells. All three groups—mid-active I and II and dead cells—showed low levels of adhesion compared to active cells. Cellular damage may alter the surface properties of probiotic cells, such as through changes in the expression or accessibility of adhesion-related molecules. For example, damage-induced modifications in the expression of lectins or surface proteins on probiotic cells may impact their ability to recognize and adhere to specific receptors on Caco-2 cells [11,13].

Cellular damage could disrupt the structures involved in probiotic cell adhesion, such as pili or fimbriae. These structures play a crucial role in facilitating adhesion to host cells. Damage-induced alterations in the structure or functionality of adhesion factors can reduce the ability of probiotic cells to adhere [14].

Damage may also play a role in the activation of various signaling pathways within probiotic cells, leading to changes in gene expression and cellular behavior. These changes can affect the expression of adhesion-related genes and proteins, influencing the adhesion capacity of probiotic cells to Caco-2 cells.

Cellular damage may cause physical damage to the outer membrane or cell wall of probiotic cells. This damage can expose inner components or disrupt the integrity of the cell surface, affecting the interaction between probiotic cells and Caco-2 cells.

Cellular damage could induce stress responses in probiotic cells, leading to the production of stress-related proteins and changes in cellular physiology. These stress responses can impact the adhesion properties of probiotic cells, potentially affecting their ability to adhere to Caco-2 cells.

Adhesion plays a crucial role in influencing the probiotic properties of bacterial cells. This is crucial for the colonization and persistence of probiotic bacteria in the gastrointestinal tract. Effective adhesion allows probiotic cells to establish a foothold in the gastrointestinal tract, which is essential for exerting their beneficial effects [15]. An important aspect of adhesion is the competitive exclusion of pathogens. Through adhesion, probiotic bacteria can competitively exclude or inhibit the attachment of pathogenic microorganisms to the intestinal epithelium. By occupying the adhesion sites on host cells, probiotic bacteria prevent the binding of pathogens, thus reducing their colonization, as well as potentially harmful effects [16,17]. Furthermore, adhesion to host cells allows probiotic bacteria to interact with other members of the gut microbiota. Through adhesion, probiotic bacteria can influence the composition and balance of the gut microbial community, promoting a beneficial microbial ecosystem [18]. The adhesion of probiotic bacteria to intestinal epithelial cells can also strengthen the epithelial barrier function. Probiotic bacteria can promote the production of tight junction proteins, strengthen cell–cell junctions, and enhance the integrity of the gut barrier. This can help prevent the translocation of pathogens or toxins across the intestinal epithelium [19]. Additionally, the adhesion of probiotic bacteria to gut epithelial cells can trigger immune responses and modulate the immune system. Probiotics can interact with immune cells present in the gut-associated lymphoid tissue, promoting beneficial immune responses and potentially regulating excessive inflammation [20]. Lastly, adherent probiotic bacteria can produce bioactive compounds near host cells. These compounds may include short-chain fatty acids [21], antimicrobial peptides [22], or metabolites, which can exert beneficial effects on the intestinal epithelium, immune system, or overall gut health.

It can be seen from these results that the bacterial cells exhibited the ability to adhere to intestinal epithelial Caco-2 cells by means of various mechanisms of cell adhesion. One such mechanism involves lectin–carbohydrate interactions, in which lectins, which are carbohydrate-binding proteins found on the surface of probiotic cells, recognize specific carbohydrates on the epithelial cell surface, thereby facilitating adhesion. For example, *Lactobacillus rhamnosus* LGG and *Lactobacillus mucosae* LM1 express lectins that interact with carbohydrates on the intestinal epithelial cells, promoting adhesion [13]. Another mechanism involves the role of surface proteins, such as adhesins, in promoting attachment. Adhesins and other surface proteins play a significant role in adhesion. These proteins on probiotic cells interact with specific receptors on Caco-2 cells. For instance, the adherence of *Lactobacillus acidophilus* to Caco-2 cells is mediated by the surface-associated protein MUB [23]. Probiotic cells can also adhere to the extracellular matrix (ECM) proteins secreted by Caco-2 cells, such as laminin or collagen. These interactions between probiotics and ECM components provide additional anchorage and stability. For example*, Lactobacillus plantarum* strains have been shown to adhere to fibronectin and mucin in the ECM [24]. Moreover, the mucus layer covering the intestinal epithelium can act as a substrate for probiotic adhesion. Mucus-binding proteins expressed on probiotic cells mediate this interaction. For instance, a comparison of patients with IBD and healthy individuals showed that some strains of *Lactobacillus* ssp. isolates exhibit weaker attachment to epithelial cells and adhere in lower numbers in the IBD patient group [25].

Cell adhesion is a complex process influenced by multiple factors. It plays a crucial role in the interaction between the probiotic cells and the gut of the host. Understating the various mechanisms involved can provide insights into the dynamics of probiotic–host interactions. One factor that impacts cell adhesion is the presence of prebiotics, such as fructooligosaccharides (FOS) or inulin. These compounds can enhance probiotic adhesion to Caco-2 cells by modifying the cell surface characteristics of both the probiotics and the Caco-2 cells. Prebiotics can promote the expression of adhesion-related molecules on probiotic cells and increase the accessibility of adhesion receptors on Caco-2 cells [26]. Gastrointestinal conditions also play a significant role in cell adhesion. Simulated gastrointestinal conditions have been shown to increase the adhesion of *Lactobacillus paracasei* to Caco-2 cells. This increase can be assigned to the increased production of extracellular polymers [27]. Temperature is another factor that influences cell adhesion. It has been reported that growth temperature affects the production of exopolysaccharides (EPS) by *Lactobacillus paracasei*, with different levels of polymerization being observed at different temperatures [28]. EPS production can impact the ability of probiotic cells to adhere to Caco-2 cells and potentially modulate their colonization and functionality. Physiological levels of shear stress in the intestinal lumen can also modulate the adhesion of probiotic cells. Under conditions of flow, the adhesion forces between probiotic cells and Caco-2 cells may differ from those under static conditions. Shear stress can influence the expression of adhesion-related genes and alter the surface characteristics of both probiotic and Caco-2 cells, ultimately affecting cell adhesion [29,30]. Host factors, including mucins, antimicrobial peptides and cytokines, can influence probiotic cell adhesion to Caco-2 cells. For instance, mucins, which are major components of the mucus layer, can affect the accessibility of adhesion receptors on Caco-2 cells and modulate the adhesion of probiotic cells. Additionally, cytokines like tumor necrosis factor-alpha (TNF-α) or interleukin-8 (IL-8) released during inflammation influence the adhesion of probiotic cells to Caco-2 cells [31]. These host factors highlight the impact of the dynamic gut environment on probiotic adhesion.

### 3.4. Adherence Patterns and Membrane Staining with DRAQ-5

IFC was also utilized to visualize the adherence of the bacterial cells after incubation with Caco-2 epithelial cells (Figure 7). DRAQ 5 was added to stain the Caco-2 cell (Ch11), and WGA was used to stain the bacterial cells (Ch02). The combined image (Ch02/Ch11) and brightfield view (Ch01) show that the bacteria adhered to the entirety of the accessible surface of the Caco-2 cell. The fluorescence channels for the detection of emitted light (Ch02 and Ch11) were selected to match the emission wavelength of selected dyes. Ch01 contains brightfield images recorded for individual events as they pass through the flow cell. 

### 3.5. The Thermal Resistance of Bacteria

Thermal stress is a primary factor, occurring during drying, influencing cell viability and probiotic properties. DSC is sensitive to protein denaturation and can be used to determine the thermal resistance of cells. Lepock [32] reported that the onset temperature of protein denaturation is correlated with growth inhibition and the onset of cell lethality. It is, however, highly strain specific, and dependent on growth conditions. For example, the onset of denaturation for *Bacillus psychrophilus* was measured at only 30 °C, while it reached 55 °C and 65 °C for *Bacillus stearothermophilus* WAT and *Bacillus stearothermophilus* ATCC12016, respectively. Figure 8 summarizes the DSC measurements of thermal stability for *E.facium* and *C. divergens* grown under optimal conditions and after heat shock. The start of the denaturation process can be observed between 65 °C and 70 °C. The exact temperatures were 68.05 °C and 66.94 °C for *E. faecium* with and without heat shock, respectively, and 68.02 °C and 64.39 °C for *C. divergens*. 

### 3.6. Analysis of Samples after Storage

The dried preparations were stored for 12 months in different atmospheres and at different temperatures. Samples were measured after 1, 2, 3, 6, 9 and 12 months of storage or until the point at which their viability decreased below the 10^6^ cfu/g threshold. The results (shown in Figure 9) show that the viability of stored preparations (measured in cfu/g) was highest for the samples stored at −20 °C, as those were the only samples to have a recorded viability of over 10^6^ cfu/g for the whole 12 months. Samples stored at 4 °C also showed an acceptable shelf-life of 6–9 months. Samples stored at 20 °C lost their viability the fastest (after only 3 months). Modified atmospheres (N_2_ and vacuum) also improved the shelf-life of samples compared to those kept in contact with air. For uncoated samples in particular, both variants had a prolonged shelf-life of 9 months compared with the 6 months for samples in contact with air. The beneficial effect of a modified atmosphere can be attributed to limiting the oxidation processes, effectively achieving the same level of stability as obtained by coating. The subpopulations in the samples were also determined using IFC, both fresh after drying and after 12 months. Most notable was the reduction in the number of active cells and the increase in both dead and mid-active subpopulations (Figure 10).

## 4. Conclusions

In conclusion, the cells that sustained membrane damage were divided into three groups (mid-active I, mid-active II, and dead) using IFC on the basis of the severity of the damage. A fourth group, containing cells with no recognizable damage, was determined and labeled as active. Only active cells exhibited the ability to adhere to the Caco-2 epithelial cell line, as confirmed by IFC. These results show that any, even minimal, damage to the cell membrane negatively affects the ability of the cell to adhere to the epithelium. This occurs even when damage to the cell membrane does not lead to cell death and the cell retains high enzymatic activity and is able to regenerate enough to maintain the ability to divide cells. DSC analysis of cell denaturation was used to determine the temperature at which denaturation started in our samples, which was between 65 and 70 °C. For dried preparations, the thermophysical analyses showed that the coating material with the highest T_g_ was HPMC, and that the samples stored at −20 °C and under a modified atmosphere had the longest shelf-life, at 12 months. Samples stored at −4 °C had a shelf-life of 6–9 months, while also being easier to store, potentially making this a better choice for commercial purposes. A comparison of the results of the IFC and pour plate counts also showed that the mid-active I and II subpopulations could not be recognized using the pour plate method. Cells with intermediate activity and low levels of cellular damage can still be viable in terms of their probiotic properties and ability to confer beneficial health effects. Commonly used classical microbiological methods are, however, unable to detect this important subpopulation, which could be included in viable cell counts for the assessment of probiotics.

## Figures and Tables

**Figure 1 nutrients-15-03484-f001:**
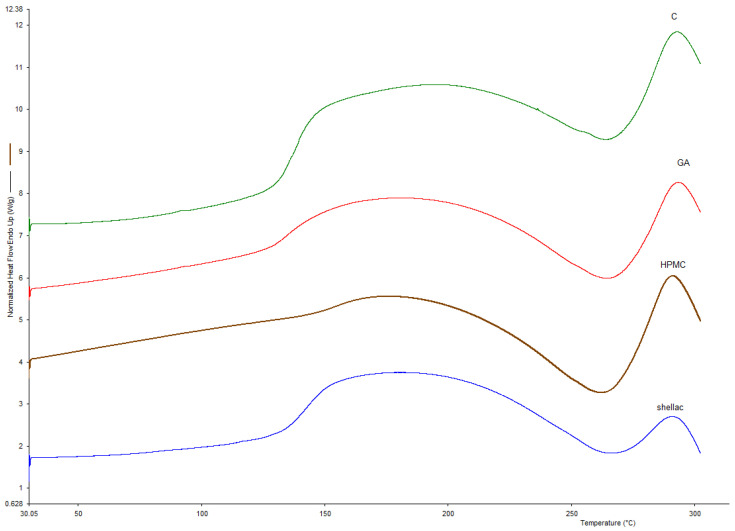
DSC graph of probiotic preparations with different coatings. C—dried, uncoated sample; GA—sample coated with gum arabic; HPMC—sample coated with hydroxypropyl methylcellulose; shellac—sample coated with shellac.

**Figure 2 nutrients-15-03484-f002:**
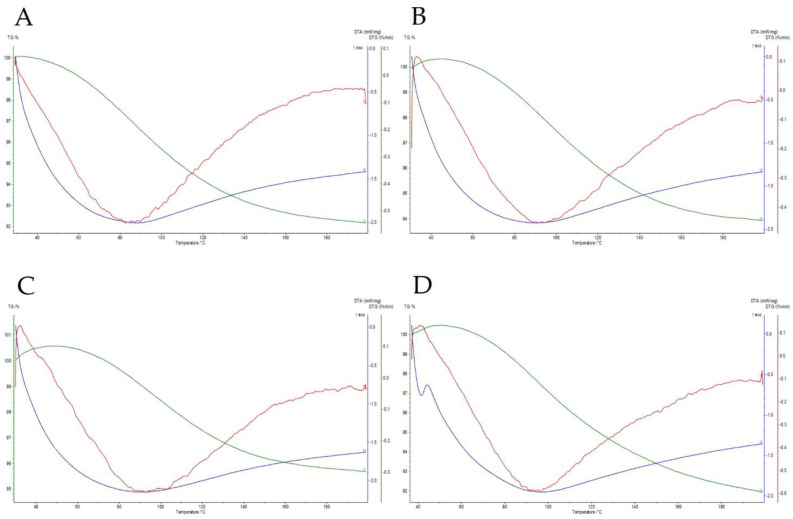
TG/DTA for (**A**) control (dried, uncoated); (**B**) samples coated with gum arabic; (**C**) samples coated with hydroxypropyl-methylcellulose; (**D**) samples coated with shellac.

**Figure 3 nutrients-15-03484-f003:**
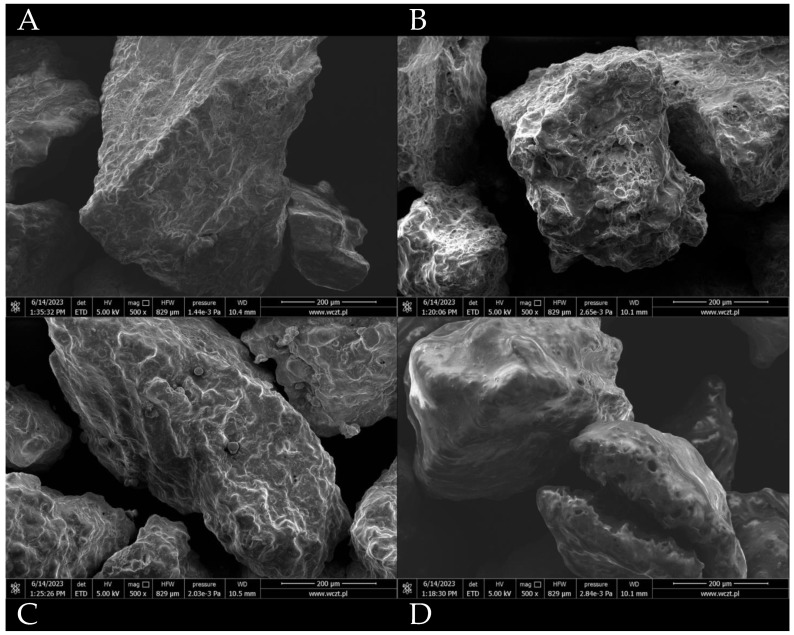
SEM pictures of dried probiotics coated with different materials. (**A**) Uncoated, (**B**) gum Arabic, (**C**) hydroxypropyl methylcellulose, (**D**) shellac.

**Figure 4 nutrients-15-03484-f004:**
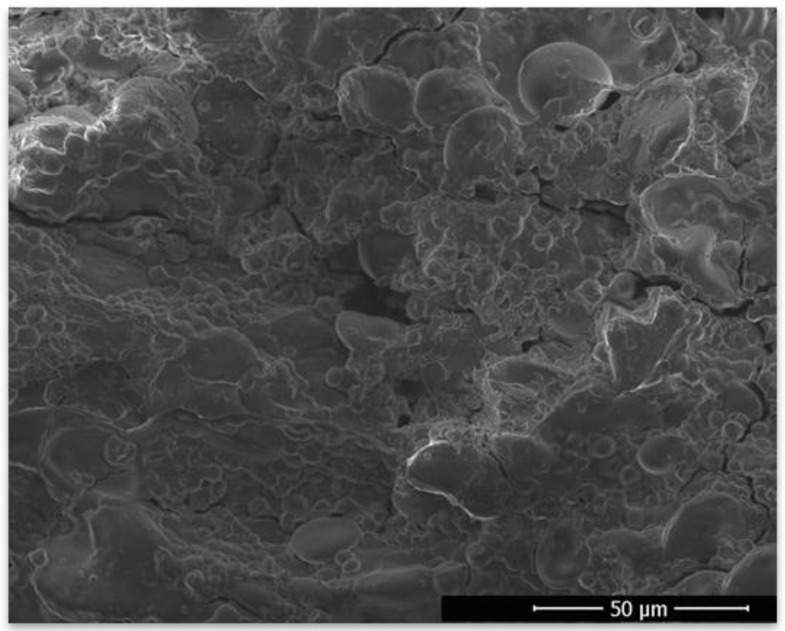
SEM picture of the surface of microcellulose matrix after drying with visible bacterial cells on its surface.

**Figure 5 nutrients-15-03484-f005:**
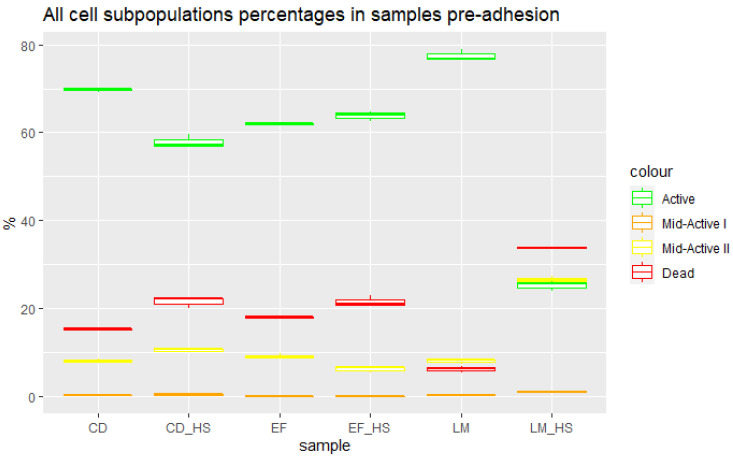
Percentages of cell subpopulations in samples pre adhesion. CD—*C. divergens*; CD_HS—*C. divergens* after heat shock; EF—*E. faecium*; EF_HS—*E. faecium* after heat shock; LM—*L. mesenteroides*; LM_HS—*L. mesenteroides* after heat shock.

**Figure 6 nutrients-15-03484-f006:**
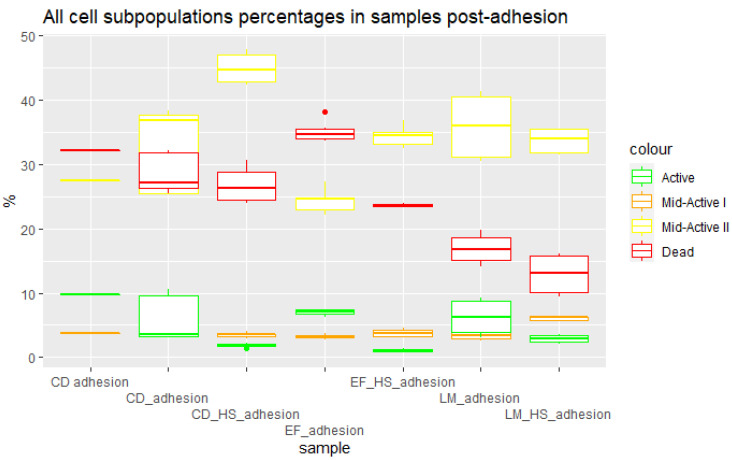
Percentages of cell subpopulations in samples post adhesion. CD—*C. divergens*; CD_HS—*C. divergens* after heat shock; EF—*E. faecium*; EF_HS—*E. faecium* after heat shock; LM—*L.mesenteroides*; LM_HS—*L. mesenteroides* after heat shock.

**Figure 7 nutrients-15-03484-f007:**
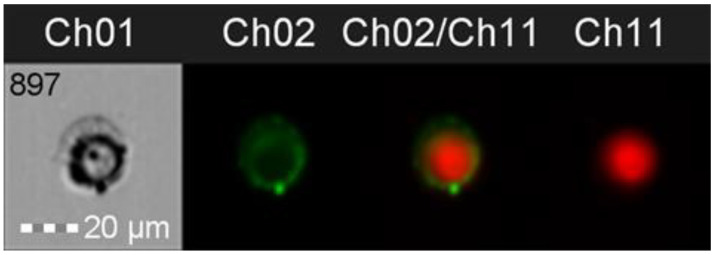
Example of adherence pattern in *E. faecium* visualized using DRAQ5 and WGA membrane dyes.

**Figure 8 nutrients-15-03484-f008:**
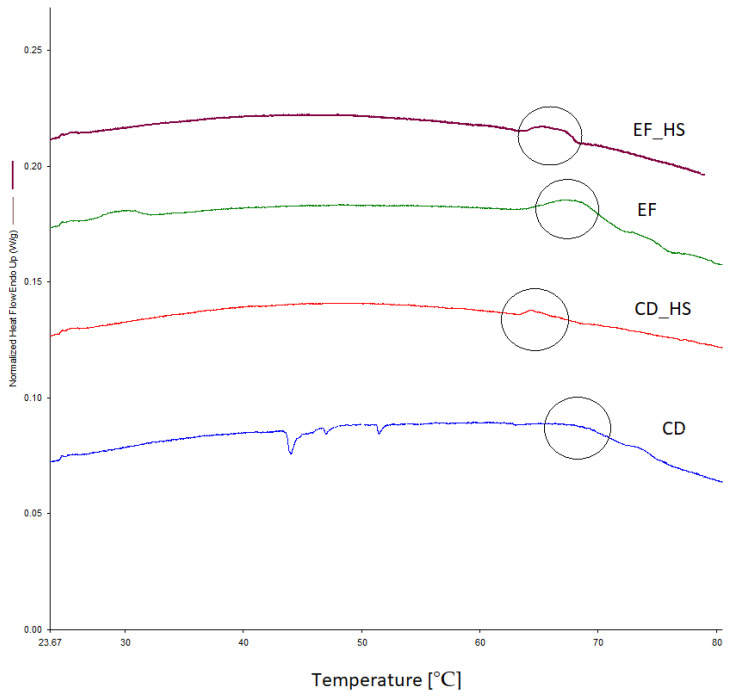
DSC graph of thermal stability for *E. faecium* and *C. divergens* cultured under optimal conditions and after heat shock. The curves for denaturation are circled.

**Figure 9 nutrients-15-03484-f009:**
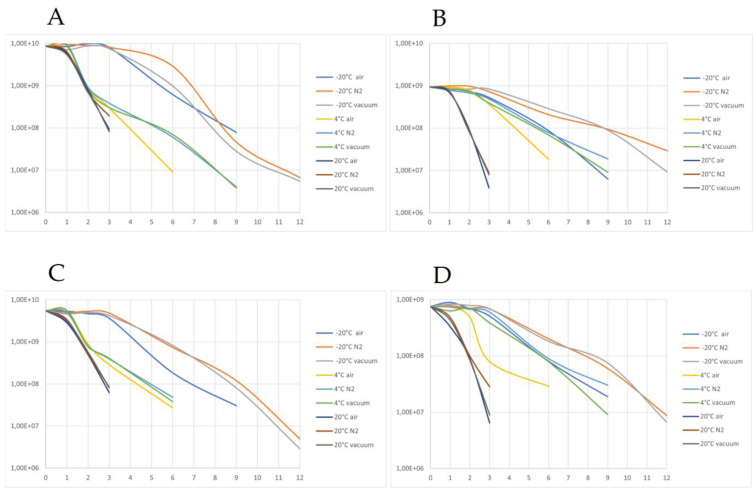
(**A**) *E. faecium* viability after storage under different conditions (−20 °C, 4 °C, 20° and packaging atmosphere with air, N2, and vacuum) for dried samples; (**B**) *E. faecium* viability after storage under different conditions (−20 °C, 4 °C, 20° and packaging atmosphere with air, N2, and vacuum) for coated samples; (**C**) *L. mesenteroides* viability after storage under different conditions (−20 °C, 4 °C, 20° and packaging atmosphere with air, N2, and vacuum) for dried samples; (**D**) *E. faecium* viability after storage under different conditions (−20 °C, 4 °C, 20° and packaging atmosphere with air, N2, and vacuum) for coated samples.

**Figure 10 nutrients-15-03484-f010:**
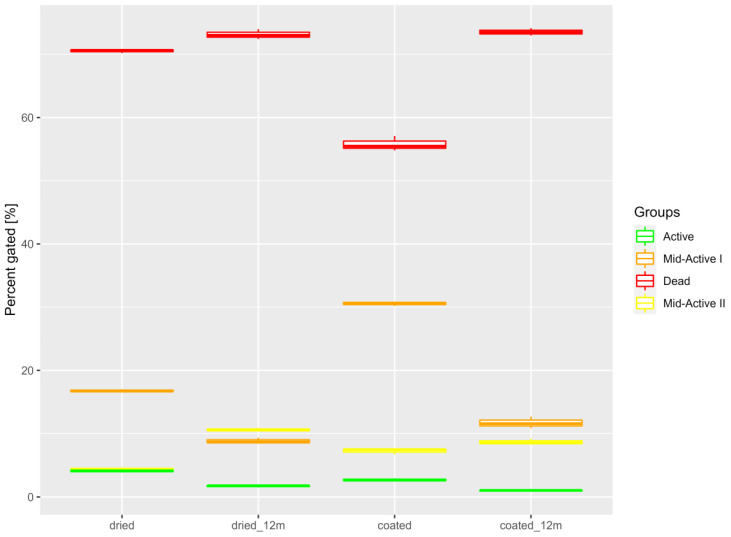
Comparison of percentages of cell subpopulations in dried and coated samples directly after processing and after 12 months of storage.

**Table 1 nutrients-15-03484-t001:** Comparison of the temperature of glass transition (T_g_) in samples coated with different materials measured using DSC and TG/DTA.

Sample	T_g_ Measured by DSC	T_g_ Measured by TG/DTA
uncoated	145.3 °C	141.3 °C
Gum arabic	144.8 °C	156.2 °C
HPMC	152.6 °C	156.1 °C
shellac	147.6 °C	140.3 °C

**Table 2 nutrients-15-03484-t002:** Enumeration of bacteria adhering to Caco-2 cells using the pour plate method.

Sample	Number of Cells Pre-adhesion [cfu/mL]	Adhered Cells [cfu/mL]	Adhered Cells [%]
*L. mesenteroides* dried	5.75 × 10^8^ + SD	4.68 × 10^7^	8.14
*L. mesenteroides* coated	5.20 × 10^8^	2.88 × 10^7^	5.54
*E. faecium* dried	9.25 × 10^8^	3.16 × 10^8^	34.12
*E. faecium* coated	6.55 × 10^8^	1.50 × 10^8^	22.96
*C. divergens* dried	4.00 × 10^8^	3.64 × 10^7^	9.10
*C. divergens* coated	1.5 × 10^8^	1.96 × 10^7^	13.07

## Data Availability

Data is contained within the article and Appendix A.

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
