# Peer review of "The Effects of Cellular Membrane Damage on the Long-Term Storage and Adhesion of Probiotic Bacteria in Caco-2 Cell Line"

_nutrients, 2023, doi:10.3390/nu15153484_

Round 1
Reviewer 1 Report
This study aims to investigate the impact of culture conditions and drying processes on the survival, storage stability, and physiological characteristics (especially the adhesion properties) of probiotic bacteria cells. This is an interesting study. I only have a few questions.
1. This study did not adequately assess the mechanisms by which cell membrane damage affects the adhesive capacity of probiotics. Furthermore, what is the significance of assessing the subpopulations with no or low levels of cell membrane damage on the adhesion of Caco-2 cells? This is because probiotics must be alive and have a good adhesion capacity. Is it meaningful to assess the adhesion capacity of those cells with low activity or even dead cells?
2. I think the title of this article is not well written, and I suggest modification.
3. line 19, I think that the statement "Cellular damage is one of the key aspects determining adhesion." is not a rigorous statement, since it is the cell membrane protein component that determines the adhesion ability of probiotics.
4. line 159, “ the PET membranes (Millicell® Cell Culture Inserts, 24 mm diameter, 0.4 m pore size)”, 0.4 m pore size?
5. There are numerous writing errors in this thesis, such as, lines 138, 130, 158, 160, 163, 179 …….
6. The figures are too loose and a proper combination is recommended.
Reviewer 2 Report
This article is covering some important aspects of the cellular membrane damage affecting the long-term storage and adhesion of probiotic bacteria in Caco2-cell lines. The title of the manuscript in the form of a question is referring to the adhesion as one of the important factors of the probiotic properties of bacteria in the human gut.
Article effectively delivers an answer for this important question highlighting cellular damage aspects determining adhesion and measuring its stability in optimal condition and treated with heat shock. Interestingly, the temperature of protein denaturation at 65o C and at 70o C as measured by DSC analysis is established as starting optimal point of their stability. The shelf life of at -4o C is determined for 6-9 months. Importantly, cells with intermediate activity and with the low level of cellular damage can still be viable in their probiotic properties to keep beneficial health effects. All this presented data constituted the important goals and novelty of this paper.
The article is concluded with a collection of 29 mostly recent references. Additionally, all 16 figures and important table with measurement data of six samples of various bacteria are critically important to quality of this important paper.
The following suggested changes and recommendations should be introduced before the publication of the manuscript.
1. Page 3. Line 136. Insert “analyzed” in front of samples.
2. Page 4. Line 186. Insert “such as” after parameters and list them accordingly.
3. Page 5. Line 251. Replace “state” with “form” after “rubbery”.
4. Page 5. Line 268. Move Table 1, behind figure 1 for better flow of information.
5. Page 11. Line 364. Table 2, move the table to the page 10 line 312 for better flow of information.
The manuscript is of good quality and importance, and is well written and edited in order to meet the standard for the articles published inNutrients. Authors should be complimented for the high quality of all figures to illustrate the determined stability under different conditions. Thus, I recommend it for proceed after the correction of these suggested minor changes and recommendations.
Reviewer 3 Report
The article shows the analysis of preparations containing probiotic bacteria, which were analyzed for temperature stability and shelf life under different conditions of pressure and refrigeration to study the adhesion of these probiotic bacteria in the human intestine. The article is well structured, and the methodology is by the intended study. However, the introduction should be improved to make the article a single well-founded unit. So my observation before proceeding this article follows below:
1) the article needs to improve the introduction by adding the importance of probiotics and the study process.
Round 2
Reviewer 1 Report
The authors have carefully answered my confusion while carefully revising the manuscript.